# Behaviour of Microwave-Heated Al_4_SiC_4_ at 2.45 GHz

**DOI:** 10.3390/ma14174878

**Published:** 2021-08-27

**Authors:** Takashi Fujii, Akio Oshita, Keiichiro Kashimura

**Affiliations:** Faculty of Engineering, Chubu University, 1200 Matsumoto-cho, Kasugai 487-8501, Japan; fujii@cs.chubu.ac.jp (T.F.); oshita.akio@gmail.com (A.O.)

**Keywords:** microwave heating, microwave absorbers, Al_4_SiC_4_, electrical permittivity

## Abstract

The ongoing development of high-temperature processes with the use of microwaves requires new microwave absorbers that are useful at these temperatures. In this study, we propose Al_4_SiC_4_ powders as important and efficient microwave absorbers. We investigated both the behavioural microwave heating and electrical permittivity characteristics of Al_4_SiC_4_ powders with various particle sizes at 2.45 GHz. The TE_103_ single-mode cavity indicated that Al_4_SiC_4_ powder samples yielded different heating behaviours and dielectric constants for each particle size compared with SiC. By microwave heating ∅50 mm × 5 mm disks of Al_4_SiC_4_ and SiC, we demonstrate that for specific sizes, Al_4_SiC_4_ can be heated at a higher temperature than SiC. Finally, the results of the two-dimensional two-colour thermometer show that an energy concentration appears at the interface of the microwave-heated Al_4_SiC_4_. These phenomena, which are inconsistent in individual physical property values, can be explained without contradicting microwave heating physics.

## 1. Introduction

High-temperature processes employing microwave heating have not been demonstrated by many researchers, despite the report by Roy et al. on microwave-based metal sintering [1]. Huang and Ishizaki et al. applied microwave heating to the steel reaction to obtain pig iron with low impurities at temperatures ≥1400 °C [2,3]. Microwave heating has been applied to the development of the hydrogen reduction method and the high-power irradiation method for iron ore [4,5]. Leonelli et al. used microwaves to provide heat for high-temperature processes in order to detoxify asbestos-containing materials [6,7]. Given that such materials have high-heat insulation properties, they can be heated more rapidly by employing microwave heating than conventional heating methods. By using microwaves to heat the inner parts of the refractory to attain high temperatures [8], the microwaves rapidly detoxify the asbestos fibres. In the future, microwave heating technology will continue to expand into the high-temperature-processing region.

To efficiently utilise the characteristics of microwaves in heating processes, materials that convert microwave energy into heat must be developed. When microwaves are used as a heat source for chemical synthesis, the raw material is heated, but the surrounding atmosphere remains cold. This feature is advantageous in chemical processes, such as amorphisation and vitrification, but for many other processes, it has detrimental effects, including decreased process efficiency and increased defective product rates. To prevent these adverse effects and take advantage of the positive effects, good microwave absorbers, such as silicon carbide (SiC) and ferrite, are placed near the material to be heated for many chemical processes [2,3,6,7]. Therefore, various researchers have developed microwave well absorbers [9,10,11,12,13].

Here, in order to evaluate the performance of Al_4_SiC_4_ as a microwave absorber for high-temperature processes, the electrical permittivity and microwave heating behaviour of Al_4_SiC_4_ were investigated. Materials used as microwave well absorbers for high-temperature operations must have high microwave absorption, high-temperature stability, and high resistance to oxidation. Al_4_SiC_4_ possesses high oxidation resistance and a thermal coefficient closer to those of Al_2_O_3_ than to those of SiC [14,15,16]. There is a trade-off between the absorption of microwaves into a substance and penetration depth. Given that Al_4_SiC_4_ does not absorb microwaves as effectively as SiC, microwaves can penetrate more deeply even at high temperatures. Therefore, because Al_4_SiC_4_ can be used to absorb microwaves that would be blocked by SiC, the substance to be heated can be directly heated by microwaves at high temperatures. If Al_4_SiC_4_ indeed possesses appropriate microwave absorption, it would be a promising candidate as a microwave absorber for use at high temperatures with Al_2_O_3_ as it could transmit at those temperatures. We first investigated both the microwave heating behaviour and electrical permittivity of Al_4_SiC_4_ powders with various particle sizes at the frequency of 2.45 GHz. Sintered Al_4_SiC_4_ was then placed in a multimode cavity similar to that used in furnaces and heated by microwaves at 2.45 GHz. This study thereby demonstrated that at a certain size Al_4_SiC_4_, can be heated to a higher temperature than SiC of the same size. Finally, the heating behaviour and deterioration of Al_4_SiC_4_ were investigated using a two-dimensional two-colour thermometer. The results showed that Al_4_SiC_4_ is an efficient microwave absorber.

## 2. Materials and Methods

### Heating Behaviour and Electrical Permittivity of Al_4_SiC_4_ Powders

To investigate their heating behaviour, Al_4_SiC_4_ powder samples (weight = 1.0 g, Tateho Chemical Industries Co., Ltd., Akou, Hyogo, Japan) with different particle radii (1 μm, 3 μm, 10.4 μm, 16.3 μm, and 1–2 mm) were heated by microwave irradiation from a magnetron oscillator at 2.45 GHz. The heating system consisted of waveguides (109.1 mm × 54.6 mm × 220.5 mm ± 5 mm) with a magnetron oscillator, an E-H tuner, a plunger, and an isolator. The microwaves were focused by an iris and formed a TE_103_ wave in this cavity. The iris had a slit with a width of 52 mm and was parallel to the direction of the electric field. The plunger was placed at the end of the waveguide. This system enabled us to spatially separate the electric and magnetic fields of the microwaves [17]. The sample was placed at an electric field node (denoted by *E*_max_) at which the magnetic field was zero. The temperature of the reactants was monitored using a radiation thermometer with a lower limit of 300 °C (FTZ9-P300-20K21, Japan Sensor Corporation, Tokyo, Japan). In these experiments, dry air flowed at a rate of 0.6 l min^−1^, and the absorption microwave power was set at 200 W. These parameters were set so that the temperature of Al_4_SiC_4_ of each particle size was constant at approximately 400–1200 °C. The irradiation time was set to 10 min to provide sufficient time for the temperature of the heated materials to stabilize to a constant value.

The real and imaginary parts of the relative permittivity (*ε**_r_*′ and *ε**_r_*″) of the Al_4_SiC_4_ powders with various particle radii (1 μm, 3 μm, 10.4 μm, and 16.3 μm) were measured to evaluate the heating mechanism. In pre-experiments, particle radii were investigated by laser diffraction (SALD-2300, Shimazu Co. Ltd., Tokyo, Japan), as shown in Figure 1. Herein, the refractive index was 1.95, and the solvent was ultrapure water. We used the cavity perturbation method [18,19] to obtain *ε**_r_*′ and *ε**_r_*″ The measurement system consisted of a cylindrical cavity and a network analyser (8753D, Keysight Co. Ltd., Tokyo, Japan). When samples were moved in the cavity, they disturbed the *Q* factor and resonant frequency of the cavity because of their material properties (i.e., *ε**_r_*′ and *ε**_r_*″). The network analyser monitored the *Q* factor and resonant frequency of the system, and *ε**_r_*′ and *ε**_r_*″ were calculated based on these values. The cavity generated the TM_020_ mode (in the TM mode, the Poynting vectors of the electromagnetic field are vertically controlled by the magnetic field). The perturbation coefficient for the estimation was assumed to be 4.318, which was assessed using a cavity without a sample. Here, the perturbation coefficient was calculated using the already measured dielectric constants of quartz and alumina rods.

The microwave heating behaviour was investigated with a multimode applicator (Nihon Koshuha, Kanagawa, Japan, maximum power = 6 kW, wave frequency = 2.45 GHz) with a maximum rated power of 6 kW. This multimode heating device was developed by Yoshikawa et al. [20], and the microwave heating device consisted of a hexagonal container. Therefore, this microwave heating furnace enables the formation of an appropriate electric field distribution inside it (Q factor = 10–500), and it can heat various materials, as shown in Figure 2. Sintered Al_4_SiC_4_ and SiC disks with specific sizes and densities (∅30 mm × 5 mm, ∅50 mm × 5 mm, density: 99%) were placed in the centre of the furnace and irradiated with microwaves at 800 W and 3.2 kW, and their temperatures were measured (atmosphere: 10 L/min dry air). A radiation thermometer (FTZ6, Japan Sensor, Tokyo, Japan) was employed at high temperatures, whereas a fibre thermometer (FL-2000, FS100-2M—190–260 °C, Anritsu, Kanagawa, Japan) was used at low temperatures (0–260 °C).

## 3. Results and Discussion

The Al_4_SiC_4_ powders show adequate microwave absorption characteristics at most particle sizes. In this measurement, to observe the electric field absorption of Al_4_SiC_4_ for its heating behaviour, a TE_103_ single-mode cavity which can heat materials depending on their electrical permittivity was used as the heating device. Figure 3 shows the time vs. temperature plots of 1.0 g of Al_4_SiC_4_ with various particle radii (1 μm, 3 μm, 10.4 μm, 16.3 μm, and 1–2 mm) heated by a TE_103_ single-mode cavity. Although the microwave power was set to 200 W in all experiments, the maximum temperature increased at all particle radii except for 3 μm. When the Al_4_SiC_4_ powders were irradiated, their temperature increased rapidly to 400–500 °C. The time vs. temperature curve shows an inflection point within this range and has a gentle slope at the temperature of the inflection point or higher. The Al_4_SiC_4_ powders with radii of 1–2 mm enhanced plasma ignition with high reproducibility (N = 3).

Figure 4 indicates the variations in temperature vs. the real and imaginary parts of the relative electrical permittivity of Al_4_SiC_4_ samples at various particle radii (1, 3, 10.4, 30.3, and 100 μm). In this case, the cavity was TM_020_, the atmosphere was 6N-grade N_2_, and their densities ranged from 1.0 to 1.8 g/cm^3^. The permittivity was measured in an inert gas to rule out the effects of the oxidation of Al_4_SiC_4_, although powders with radii of 1–2 mm could not be loaded into the instrument and could not be measured. As shown, the real part of the dielectric constant increased as the temperature increased. At room temperature (25 °C), the real part of the dielectric constant was high and followed the order of 30.3 μm > 3 μm > 10.4 μm > 1 μm, but as the temperature of the Al_4_SiC_4_ increased, the order changed to 3 μm > 1 μm > 30.3 μm > 10.4 μm > 100 μm. Similarly, the imaginary part of the dielectric constant increased with the increasing temperature. At room temperature, the real part of the dielectric constant was also high, but it followed the order of 100 μm > 10.4 μm = 30.3 μm > 3 μm. As the temperature of Al_4_SiC_4_ increased, the order changed to 3 μm > 1 μm = 30.3 μm > 10.4 μm.

The Al_4_SiC_4_ powder samples with diameters of 3 μm could not be easily heated by microwaves with high accuracy. The temperature of a material heated by microwaves is determined by the difference between the microwave absorption energy in the material and the thermal energy that is lost from the materials. Given that the microwave absorption energy W is proportional to the imaginary part of the relative permittivity *ε*″_*r*_ ($w\sim ∰\frac{1}{2}\omega \varepsilon_{0}\varepsilon_{r}^{\prime \prime }{\left|E\right|}^{2}dV$ at *E*_max_, where $\varepsilon_{0}$: electrical permittivity, *E*: strength of electrical field, and ω: angular velocity), the imaginary part increases with temperature, as shown in Figure 4.

Notably, the imaginary part of the electrical permittivity of Al_4_SiC_4_ is an order of magnitude lower than that of SiC reported in previous studies [12,13]. This value is an index which quantifies how microwave energy is absorbed per unit material volume. The higher this value is, the easier the material can absorb microwaves [8,13]. However, as this value increases with increasing temperature, it becomes more difficult for microwaves to penetrate deeply into the material. Thus, the ratio between the dielectric constants of the material and air becomes larger, and microwaves are more easily reflected by the material. Therefore, SiC and carbon, which are known as microwave well absorbers, act as microwave reflectors in processes at temperatures ≥600 °C [12,13]. However, the electrical permittivity (imaginary part) of Al_4_SiC_4_ remains constant at high temperatures; thus, Al_4_SiC_4_ can be expected to behave as an absorber even at high temperatures.

Thus, at high temperatures and certain particle sizes, Al_4_SiC_4_ is superior to SiC as a microwave absorber, possibly because microwaves can penetrate further into a material with moderate microwave absorption properties. Figure 5a shows time vs. temperature plots of sintered Al_4_SiC_4_ samples (∅50 mm × 5 mm, ∅30 mm × 5 [mm]) irradiated by microwaves at 800 W, where the percentage means the purity of the sample. The temperatures of the sintered Al_4_SiC_4_ and SiC were measured by a fibre thermometer. During the microwave heating of the Al_4_SiC_4_ and SiC disk (∅50 mm × 5 mm) shown in Figure 5a, it was necessary to turn off the microwave output to avoid damaging the thermometer as the sample approached 240 °C. Therefore, the irradiation time was decided accordingly. A preliminary test confirmed that an alumina disk of the same size was not heated by microwaves. A sintered Al_4_SiC_4_ disk with a diameter of 30 mm exhibited a temperature rise in the range of 50–100 °C under microwave irradiation, and the heating rate of sintered Al_4_SiC_4_ depended on the density. By contrast, the sintered Al_4_SiC_4_ disk with a diameter of 50 mm heated to a higher temperature than a disk with an equivalent density but a diameter of 30 mm. Considering that the sintered Al_4_SiC_4_ (∅50 mm × 5 mm) density is relatively close to that of the sintered Al_4_SiC_4_ (∅30 mm × 5 mm) disk, this finding implies that the disk size affects the microwave heating behaviour. Notably, the heating rate of the sintered SiC disk was higher than that of sintered Al_4_SiC_4_ disk, which indicates that SiC exhibits higher microwave absorption than Al_4_SiC_4_ at low temperatures. However, as shown in Figure 5b, the 3.2 kW microwaves heated the sintered Al_4_SiC_4_ to approximately 200 °C, which was higher than that of the sintered SiC. Further, at higher temperatures, Al_4_SiC_4_ is heated more effectively than SiC, which indicates that Al_4_SiC_4_ may be superior to SiC as a microwave absorber in the high-temperature region.

Sintered Al_4_SiC_4_ compounds behave as a better microwave absorber than SiC at temperatures >600 °C for two reasons. One is the size of sintered Al_4_SiC_4_. A material with good microwave absorption rapidly absorbs microwave energy. Thus, as the size increases, the microwaves do not heat the interior but heats only the surface. Conversely, in a material with moderate microwave absorption properties, microwaves can penetrate further into the interior. Therefore, most of the microwave energy can be absorbed even if the imaginary part of the relative electrical permittivity is small. Thus, as the size increases, SiC loses this advantage in microwave heating. The other reason is that SiC exhibits a high microwave reflection response at high temperatures. High-temperature SiC has many free electrons on the Fermi surface, so it behaves metallically toward microwaves [12,13]. Therefore, graphite and SiC reflect microwaves at high temperatures. For these two reasons, it is speculated that sintered Al_4_SiC_4_ outperforms SiC as a high-temperature microwave absorber. In addition, Al_4_SiC_4_ has a coefficient of thermal expansion similar to that of Al_2_O_3_ [14], which is often used as a refractory material in high-temperature processes. This is another important reason why Al_4_SiC_4_ would be a useful candidate for microwave absorbers in such processes.

Microwave heating of Al_4_SiC_4_ in air causes another interesting phenomenon at the particle interfaces. Figure 6 shows the time vs. temperature plots of Al_4_SiC_4_ particles (2 mm) during microwave heating in a 100 W electric field, along the two-dimensional two-colour thermometer measurement results. As shown in this figure, when Al_4_SiC_4_ is heated in air, the particle surface temperature becomes approximately 200 °C higher than that in the interior. This temperature difference can possibly be attributed to the formation of oxide films on the Al_4_SiC_4_ surfaces when it is heated in air. Thus, these oxide films exhibit different heating responses. Interestingly, this interfacial gradient cannot be explained without assuming a thermal conductivity that is several hundred times lower than that of air according to the law of heat conduction.

Scanning electron microprobe (SEM) and energy-dispersive X-ray (EDX) analyses of the Al_4_SiC_4_ before and after microwave heating (dry air, 1000 °C, 10 min) provide deeper insights into this interfacial gradient, as shown in Figure 7. Before microwave heating, the surface of Al_4_SiC_4_ exhibited strong Kα rays indicating C and Si, with the latter being homogeneously distributed. However, after microwave heating, white particles adhered to the surface of the Al_4_SiC_4_. In addition, the amount of carbon decreased, but the oxygen increased after heating. This result suggests that a special electric field concentration is formed by the surface coating, according to Maxwell’s formula [21].

Remarkably, more Si seemed to evaporate during heating than Al. Because Al has a lower ionisation energy than Si, Al has a higher probability of plasma ionisation in the presence of an electric field. However, the SEM analysis indicates that Si may have evaporated, which implies that the evaporation of the interfacial elements follows a thermodynamic mechanism.

## 4. Conclusions

Al_4_SiC_4_ powders were heated by microwaves at 2.45 GHz to demonstrate a novel microwave well absorber for high-temperature microwave processing. Specifically, we investigated the microwave heating behaviour and electrical permittivity characteristics of Al_4_SiC_4_ powders with various particle sizes. The results can be summarized as follows.

The Al_4_SiC_4_ powders were adequately heated by microwaves. The imaginary parts of their electrical permittivity yielded values in the range of 0.032–0.240. For Al_4_SiC_4_ particle sizes in the range of 3 μm–2 mm, larger particles led to higher maximum temperatures during microwave heating and higher imaginary parts of the electrical permittivity.In the particle size range of 1 μm–2 mm, the Al_4_SiC_4_ powders were adequately heated by microwaves as the particle size increased. This tendency was also consistent with the complex permittivity of the Al_4_SiC_4_.According to measurements by the resonance perturbation method, the higher the temperature of the Al_4_SiC_4_ powders (from 25 °C to 600 °C), the higher the real and imaginary parts of their electrical permittivities.The sintered Al_4_SiC_4_ (∅50 mm × 5 mm) disk could be heated more effectively with microwaves than a sintered SiC disk of the same size. This tendency was presumed to become more pronounced with the increasing size of the sintered body and the rising temperature.

Al_4_SiC_4_ has a coefficient of thermal expansion similar to that of Al_2_O_3_ and low electrical conductivity even at high temperatures. It is, therefore, considered to be a promising candidate as a microwave well absorber for high-temperature processes that use microwaves.

## Figures and Tables

**Figure 1 materials-14-04878-f001:**
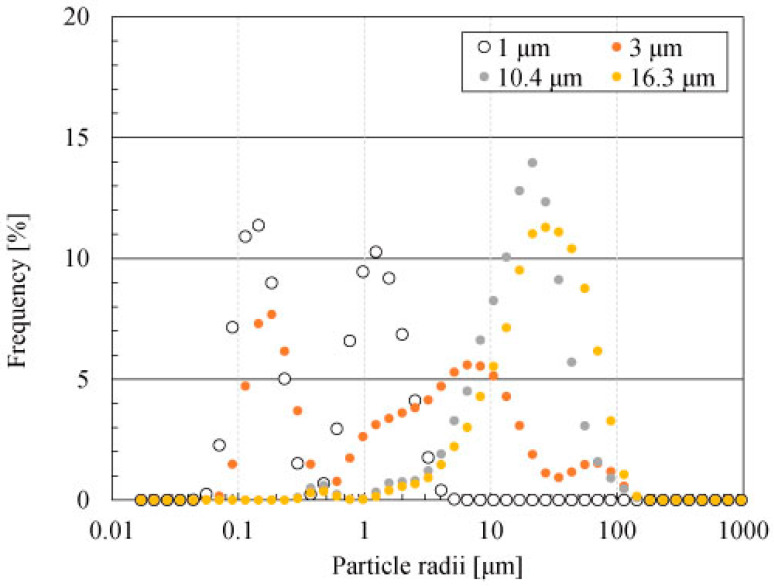
Particle radii of each powder (1 μm, 3 μm, 10.4 μm and 16.3 μm) investigated by a laser-diffraction-type particle size distribution measuring device.

**Figure 2 materials-14-04878-f002:**
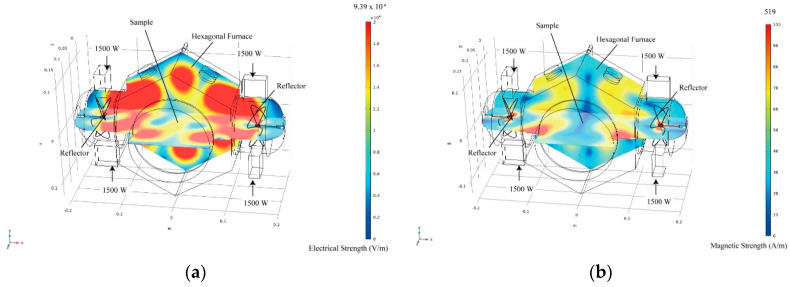
(**a**) Schematics of the experimental apparatus along with the electrical field distribution and (**b**) the magnetic field distribution at 6 kW (1500 mm × 4 mm).

**Figure 3 materials-14-04878-f003:**
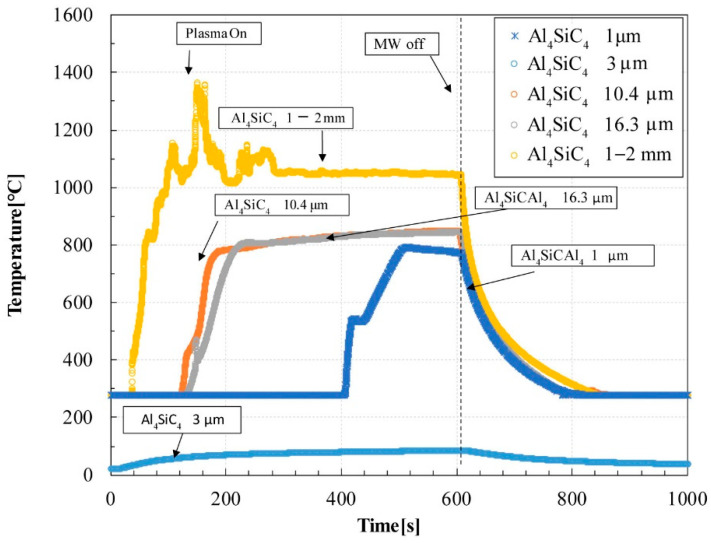
Time vs. temperature plots of Al_4_SiC_4_ samples (weight = 1.0 g) heated in a TE_103_ cavity. Microwaves generated at a rated power of 200 W were focused on the sample, which was maintained in dry air conditions (0.6 L/min flow) during the heating experiment.

**Figure 4 materials-14-04878-f004:**
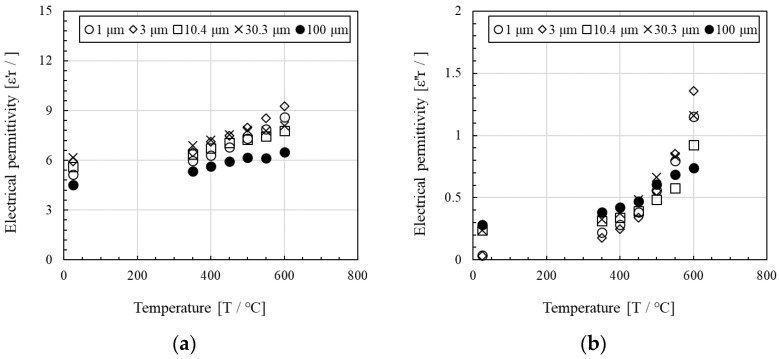
Plots of temperature vs. the (**a**) real and (**b**) imaginary relative electrical permittivity of Al_4_SiC_4_ samples with various particle radii (1, 3, 10.4, 30.3, and 100 μm). Experiments were conducted in a 6N N_2_ atmosphere, and the sample densities were 1.0–1.8 g/cm^3^.

**Figure 5 materials-14-04878-f005:**
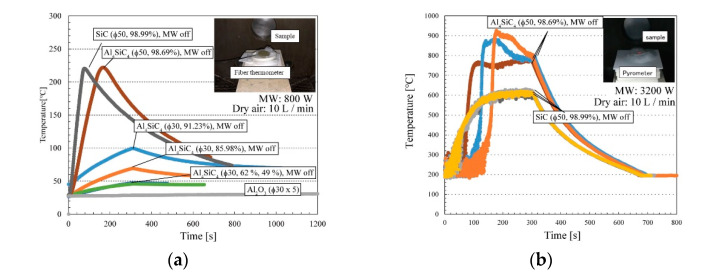
(**a**) Time vs. temperature plots of sintered Al_4_SiC_4_ samples (∅50 mm × 5 mm, ∅30 mm × 5 mm) irradiated with microwaves at 800 W and (**b**) 3.2 kW. Preliminary tests confirmed that the alumina and refractory bricks used for the table were not heated by the microwaves.

**Figure 6 materials-14-04878-f006:**
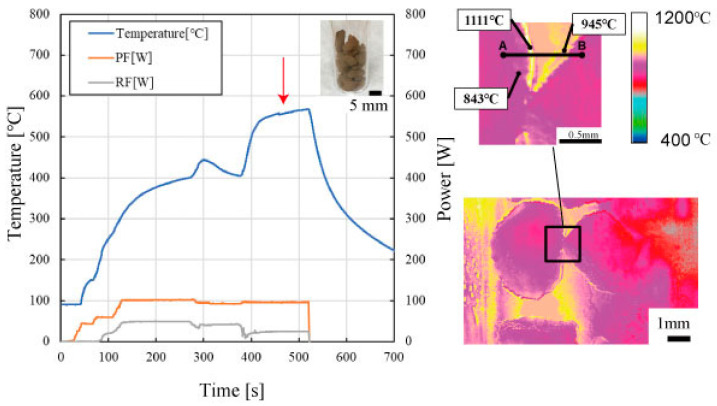
Time vs. temperature plots of Al_4_SiC_4_ particles (2 mm) during microwave electric field heating at 200 W and two-dimensional two-colour thermometer measurement results.

**Figure 7 materials-14-04878-f007:**
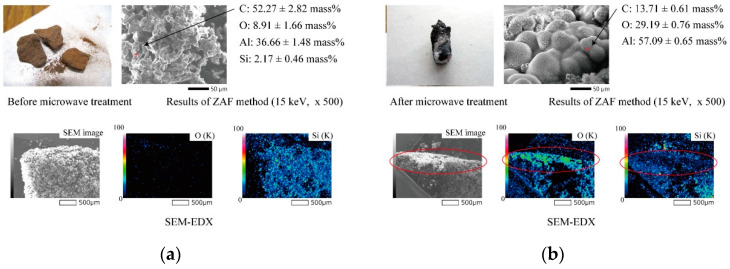
SEM-EDX images and analysis of Al_4_SiC_4_ (**a**) before and (**b**) after microwave heating (dry air, 1000 °C, 10 min).

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
