# Peer review of "Behaviour of Microwave-Heated Al4SiC4 at 2.45 GHz"

_materials, 2021, doi:10.3390/ma14174878_

Round 1

Reviewer 1 Report

Review

Comments:

  1. Line 19, section Abstract states that “Al4SiC4 has energy concentration observed at the interface”. This was to be expected due to the special dielectric properties of Al2O3, which should be coating the analyzed particles.
  2. Lines 178-179: "It has been confirmed in a preliminary test that an alumina disk of the same 178 size is not heated by microwaves." However, this coating of particles can shield them from microwave exposure and distort results. This point should be discussed in the article.
  3. Lines 220-221 “One is that 220 Al4SiC4 forms an oxide film on its surface when heated in air. Thus, it is possible that these oxide films exhibit specific heating responses." Then it is not clear why the article lacks data of transmission electron microscopy, which could help to show the presence of particles of the core-shell type, which is most likely in this study. These data also could help to determine the thickness of the Al2O3 shell and adjust the interpretation of the results accordingly.

This is supported by the results shown in lines 197-198 and 208. After making the specified additions and clarifications, the article can be published.

Author Response

Response for Reviewer #1

We deeply appreciate the theoretical proposals of the referees while reviewing our manuscript. They have improved the quality and presentation of our paper. We agree with the comments and have revised the manuscript accordingly. All revisions made to the text based on the reviewer’s suggestions are shown in red in the revised manuscript.

We hope that the revised paper is suitable for publication. Should you require any clarifications, please contact me.

Thank you for your consideration.

Sincerely,

Keiichiro Kashimura

********************************************************************************

Suggestion 1: Lines 178-179: "It has been confirmed in a preliminary test that an alumina disk of the same 178 size is not heated by microwaves." However, this coating of particles can shield them from microwave exposure and distort results. This point should be discussed in the article.

Response: Thank you for your suggestion. The heating data of the alumina disk was added to Fig. 4-2.

Suggestion 2: Line 19, section Abstract states that “Al4SiC4 has energy concentration observed at the interface”. This was to be expected due to the special dielectric properties of Al2O3, which should be coating the analyzed particles.

Suggestion 3: Lines 220-221 “One is that 220 Al4SiC4 forms an oxide film on its surface when heated in air. Thus, it is possible that these oxide films exhibit specific heating responses.

Suggestion 4: " Then it is not clear why the article lacks data of transmission electron microscopy, which could help to show the presence of particles of the core-shell type, which is most likely in this study. These data also could help to determine the thickness of the Al2O3 shell and adjust the interpretation of the results accordingly.

Response: We recognise that these points are significant important, which give a theoretical interpretation to the experimental results. To respond the question, Al4SiC4 heated for 10 minutes was analysed by SEM. At first, we observed Al4SiC4 embedded in resin; however, the oxide film was peeled off by polishing. In order to avoid this problem, the sample not to be embedded in the resin was lightly polished, and the part where the oxide film was presumed to remain was observed by SEM. As a result, we were able to observe a decrease in oxide film and Si and C at Al4SiC4 after microwave heating. Therefore, the obtained data was added, and the consideration was revised. The main changes are as follows.

We have added Figure 7.

Lines: 243–260

We have added more references for theoretical background.

Reviewer 2 Report

The paper certainly raises an important point in the use of microwaves in industrial processes, dealing with materials used as matrices for catalysts to NOx reduction in exhaust treated in a tested experimental apparatus.

The paper is interesting and offers points to ponder, especially in the field of applied microwaves, powerful as much as complicated tools. Nevertheless, my personal suggestion is the paper be carefully revised before publication, as some points need to be better developed and explained:

  • The main goal is not clearly described, as it seems only directed to test the microwave effect on an inorganic material.
  • The selections of the experimental conditions are not clearly justified: power of irradiation, quantity and size of samples, times of irradiation, kind and flow rate of gas in the chamber, to name some. A scheme of the
  • Some discrepancy should be checked and solved (in line 88 “Powder samples with radii in the range of 1–2 mm could not be loaded in the instrument and could not be measured” whereas results are displayed in Fig. 2).
  • It is not clear the sense of the comparison (line 182-185) of discs of the different diameter of the same material.
  • If available, displaying the magnetic field in the chamber by side of the electric one shown in Fig. 1, could be very interesting.
  • Figure 4 (b) would certainly take advantage of a lesser number of experimental points
  • The puzzling as much as interesting point raised in analyzing results displayed in Fig. 5, whether the higher temperatures of the particle surface are due to the formation of an oxide film or of plasma, would perhaps benefit of experimental runs in inert atmosphere.

Author Response

Response for Reviewer #2

We deeply appreciate the efforts of the referees while reviewing our manuscript. They have improved the quality and presentation of our paper. We agree with the comments and have revised the manuscript accordingly. All made to the text based on the reviewer’s suggestions are shown in red in the revised manuscript.

We hope that the revised paper is suitable for publication. Should you require any clarifications, please contact me.

Thank you for your consideration.

Sincerely,

Keiichiro Kashimura

********************************************************************************

Suggestion 1: The selections of the experimental conditions are not clearly justified: power of irradiation, quantity and size of samples, times of irradiation, kind and flow rate of gas in the chamber, to name some.

Response: Thank you for your helpful suggestions. In order to justify each measurement condition, the explanation of the purpose of the experiment and the experimental parameters was added to the item of measurement result. We received the indication of particle size distribution as a particularly important indication, so we measured the particle size distribution by laser diffraction. The main additions are listed below.

Fig. 1

Lines: 125–127

Lines: 83–86

Lines: 89–91

Lines: 186–188

Lines: 146–149

Suggestion 2: A scheme of the Some discrepancy should be checked and solved (in line 88 “Powder samples with radii in the range of 1–2 mm could not be loaded in the instrument and could not be measured” whereas results are displayed in Fig. 2).

Response: We sincerely apologise for the ambiguity. The description pointed out was deleted and the relevant explanation was added to Fig. 3.

Lines: 140–142

Suggestion 3: It is not clear the sense of the comparison (line 182-185) of discs of the different diameter of the same material.

Response: Thank you for your pointing this out. SiC is superior to Al4SiC4 in microwave absorption. However, this superiority or inferiority is reversed when the SiC and Al4SiC4 have a high temperature and a certain size. Therefore, we heated a large size sintered Al4SiC4 to investigate the possibility that Al4SiC4 is a better than SiC as microwave absorber. In response to reviewer’s suggestions, I have added an explanation for the above points.

Lines: 180–183

Lines: 212–216

Suggestion 4: If available, displaying the magnetic field in the chamber by side of the electric one shown in Fig. 1, could be very interesting.

Response: Thank you for suggestions. We have calculated magnetic field in the chamber.

We add magnetic field distribution in the figure.

Suggestion 4: Figure 4 (b) would certainly take advantage of a lesser number of experimental points

Response: Thank you for suggestions. We have retried the experiment and obtained adequate reproducibility.
